# Participatory research towards the control of snakebite envenoming and other illnesses in a riverine community of the Western Brazilian Amazon

**Alícia Patrine Cacau Santos**[1,2], **Evellyn Antonieta Rondon Tomé Silva**[1,2], **Hiran Sátiro Souza da Gama**[1,2], **Jady Shayenne Mota Cordeiro**[1,2], **Ana Paula Silva Oliveira**[1], **Jéssica Albuquerque Araújo**[3], **Rafaela Nunes Dávila**[1,2], **Hélio Afonso Amazonas Júnior**[1], **Altair Seabra Farias**[1,2], **Jacqueline Almeida Gonçalves Sachett**[1,2], **Vinícius Azevedo Machado**[1,2], **Wuelton Marcelo Monteiro**[1,2], **Felipe Leão Gomes Murta**[1,2]*

1 Fundação de Medicina Tropical Dr. Heitor Vieira Dourado, Manaus, Brazil, 2 Universidade do Estado do Amazonas, Manaus, Brazil, 3 Fiocruz Amazônia–Instituto Leônidas & Maria Deane, Manaus, Brazil

* felipelmurta@gmail.com

**Data Availability Statement:** All relevant data are within the manuscript and its Supporting Information files.

## Abstract

### Background

Riverine communities face various health problems, which involve geographical and cultural barriers to accessing care, in addition to a lack of financial investments in services aimed at these communities, resulting in a process of invisibility for the population living in these regions. In this scenario, the significant burden of snakebite envenoming (SBE) highlights the need for participatory research to address ways to minimize this situation. Thus, this study aimed to describe the priority health problems identified by this population and the ranking of SBEs in that context, mapping solutions according to the local reality.

### Methodology/Principal findings

This study was conducted in Limeira, a riverine community located in Tabatinga, in the extreme Western Brazilian Amazonia, on the borders with Peru and Colombia. The research lasted approximately one year, from 2021 to 2022. It is a participatory study that followed three steps: baseline assessment of the community, community assembly, and final data analysis.

The study included a total of 42 participants in the sociodemographic survey, which served as the basis for the subsequent stages of data collection. Of these 42 individuals, 32 participated in the qualitative interviews, and 20 took part in the community assembly. Participants emphasized snakebite envenoming as a significant health issue, though not the only one, and reported frequent encounters with snakes, underscoring its severity as a concern. The qualitative analysis identified three main themes: Snakebites in the Community, which focused on personal experiences with snakes; Common Health Problems, which addressed other health issues faced by community members; and Community Defining

**Funding:** 1. This study was funded by Fundação de Amparo à Pesquisa do Estado do Amazonas – FAPEAM - RESOLUÇÃO N. 002/2023 - POSGRAD 2023 - Coordenador/Auxílio Financeiro, This study also was supported the Fundação de Amparo à Pesquisa do Estado do Amazonas (FAPEAM) EDITAL N. 017/2023 – PRONEM, and Fundação de Amparo à Pesquisa do Estado do Amazonas (FAPEAM) Resolução N. 023/2022 - INICIATIVA AMAZÔNIA +10 to WMM. APCS received a PhD's research scholarship from POSGRAD UEA FAPEAM. The funders had no role in study design, data collection and analysis, decision to publish, or preparation of the manuscript.

**Competing interests:** The authors have declared that no competing interests exist.

Solutions, which discussed strategies and solutions proposed by the community to address these challenges.

## Conclusions/Significance

Improvements in health care delivery to populations living in Amazonian communities are possible with the judicious use of tested integrated interventions, particularly when the community identifies various concurrent health problems. SBE control programs in remote areas of the Brazilian Amazon should be planned with a multidisciplinary and intercultural approach, preferably integrated with broader interventions that address the population's needs for a range of health issues.

### Author summary

Riverine communities in the Brazilian Amazon have been facing various health problems due to factors such as the region's geographic characteristics, lack of investment in public policies, and difficulty accessing healthcare services, among others, which result in the invisibility of this population in the country's public policies. This participatory study, conducted in Limeira, a riverine community in the interior of the Amazonas state, Brazil, reveals the various challenges faced by the community members, including tropical and infectious diseases, occupational diseases, work-related accidents, women's health issues, among others. Additionally, the study collectively mapped out strategies and solutions to address these problems, which threaten the quality of life and overall well-being of this population.

## Introduction

Snakebite envenoming (SBE) is a critical health issue for riverine communities in the Brazilian Amazon, where the combination of environmental factors, limited access to healthcare, and the unique lifestyle of these populations heightens the risks [1,2]. In the State of Amazonas, one of the areas with the highest incidence of SBE globally, up to 200 cases are reported per 100,000 inhabitants annually, and the mortality rate stands at 0.7%, with rural residents being the most affected [3]. This is compounded by the fact that timely medical care is essential; delays of over six hours in treatment can lead to higher risks of severe complications and death [4]. For these communities, the threat of SBE affects daily routines, from attending school to essential activities such as fetching water or using the restroom [5]. The serious and often debilitating consequences of SBE, such as strokes and permanent disabilities, underscore the urgent need for improved healthcare access and antivenom distribution [6,7].

The challenges posed by SBE are closely tied to the broader socio-historical context of the region. The historical social formation of riverine communities in the Brazilian Amazon is deeply associated with the process of colonization and occupation which involved a series of physical, cultural, and symbolic violences that shaped the historical context [8,9]. Especially during the initial period of the Portuguese invasion of Brazil and the migration of Brazilian northeasterners to the Amazon region during the rubber boom [10,11]. The often non-peaceful interaction between these different populations contributed to the formation of a highly mixed group, known as the riverine people [12]. The word 'ribeirinho' in Portuguese seeks to

identify the social group that established their way of life along the riverbanks, maintaining symbolic, cultural, and social connections with the environment in which they live [9]. Their way of life is dynamically connected to the environment, through family farming, fishing, hunting, traditional knowledge, and especially their relationship with the rivers, forming both emotional and survival bonds [13,14].

However, the socioeconomic development resulting from urban-industrial projects, such as hydroelectric plants and infrastructure projects, has increased inequalities and marginalization for these communities [15,16]. The logic of urban-industrial growth has often led to the exploitation or destruction of natural resources, further disenfranchising these communities and increasing their social and political invisibility [17,18].

These challenges are compounded by geographical barriers inherent to the region's remote and difficult-to-access areas, which are now being worsened by climate change and environmental degradation [19,20]. The effects of deforestation, mining, fires, and extreme weather events, such as floods and droughts, have placed even greater strain on these communities, affecting their health and livelihood [21–23]. The disruption of local ecosystems has altered the transmission dynamics of tropical diseases like malaria, dengue, and leishmaniasis, which further exacerbates the health challenges these communities face [20,24]. Additionally, these environmental changes have affected wildlife behavior, including that of venomous snakes, leading to an increase in SBE cases, which is a serious health problem in the Brazilian Amazon [1,25].

Considering these complexities, Brazil is committed to achieving the Sustainable Development Goals, particularly Goal 3—Health and Well-Being, which aims to ensure healthy lives and promote well-being for all ages. This endeavor requires a fundamental understanding of the riverine communities' perspectives on health issues, which can be achieved through participatory research [26].

Given these environmental, social, and healthcare challenges, it is crucial to understand how riverine communities perceive and experience health issues, especially in the context of SBE. The current gaps in healthcare access and the increased prevalence of health issues like SBEs in these populations highlight the need for targeted health interventions. Therefore, the objective of this study was to identify the main health challenges present in a riverine community through participatory research, to understand the role of SBE in this context, and mapping strategies to address all these health issues. We hypothesize that the health challenges faced by riverine communities, such as SBE, can be better understood within their sociocultural and environmental context through participatory research. This approach allows the identification and prioritization of health issues as perceived by the community itself, highlighting how SBE is interconnected with other significant health concerns and shaping strategies to address these challenges holistically.

## Methods

### Ethics statement

The Ethics Review Board at the Fundação de Medicina Tropical Dr. Heitor Vieira Dourado in Manaus (CAEE: 40850020.1.0000.0005) approved the study. Eligible participants provided informed consent before participation, following ethical guidelines in Brazil. For literate participants, written consent was obtained, and each participant received a copy of the consent form. For illiterate participants, verbal informed consent was recorded, accompanied by the signature of a legally authorized witness, as required by Brazilian legislation. The study adhered to the COREQ (Consolidated Criteria for Reporting Qualitative Research) guidelines [27] (S7 File) to ensure rigorous and ethical reporting of the research process and findings.

## Context of study

This report presents the initial findings of a project originally aimed at understanding SBE within a riverine community. However, during data collection, participants highlighted various other health issues that they felt needed to be integrated into the study to provide a more comprehensive and meaningful analysis. Therefore, this study not only explores the primary health problems identified by community members but also examines the role of SBE within this broader context [28]. This part of the study lasted approximately one year and took place between 2021 and 2022.

## Study area and population characteristics

This study was conducted in Limeira, a riverine community situated in the city of Tabatinga, in the western part of the state of Amazonas, Brazil. Limeira is located along the banks of the Solimões River, a vital segment of the Amazon River (Fig 1) [29].

Access to the Limeira community is generally facilitated by motorized canoes, though their availability can vary with the seasonal changes in the region. During the Amazonian winter, which spans from mid-October to June, canoes become the primary mode of transportation, as this period is marked by heavy rains and river flooding. As a result, canoes are commonly used for everyday activities such as fishing, traveling to Tabatinga city to purchase supplies, transporting children and adolescents to school, and collecting firewood, among other tasks. Conversely, during the Amazonian summer, motorcycles and walking become the principal

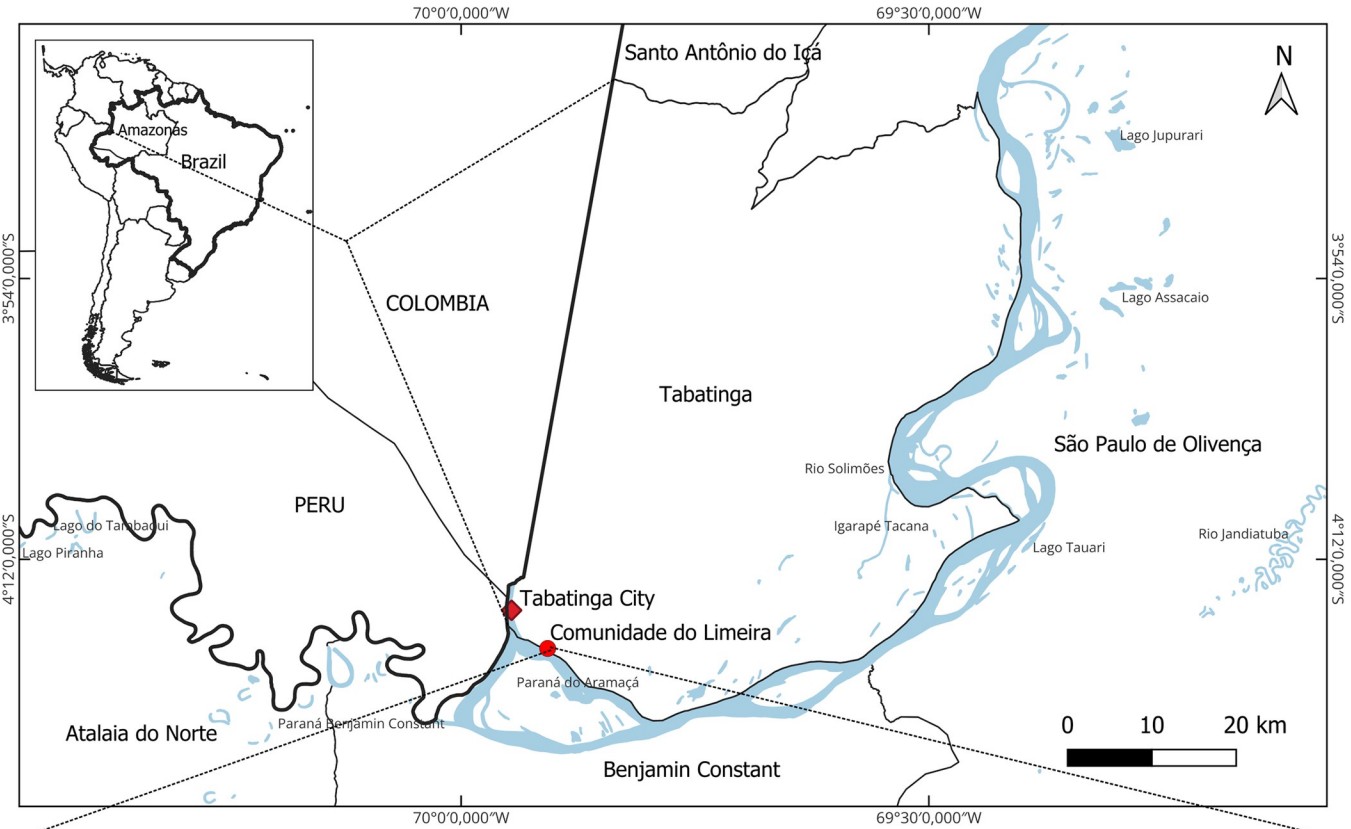

**Fig 1. Study site, Limeira community in Tabatinga, Amazonas, Brazil.** The base used to create the map is from the IBGE (Brazilian Institute of Geography and Statistics), which is freely accessible for creative uses in shapefile format, following the Brazilian Access to Information Law (12,527/2011) [30].

means of transportation, as the rivers recede, and water levels reach critical lows. This dry season, lasting from July to October, presents challenges as extensive sandbanks emerge, separating the community from the Solimões River. Additionally, the road connecting Limeira to Tabatinga is extremely rugged, lacking pavement and public lighting, and is seldom used by the riverine inhabitants.

Approximately 64 people live in the community, with the main activities being subsistence farming and fishing. A total of 42 individuals were included in the research, with 21 (50%) being women and 21 (50%) men. The median age for women was 30 years, while for men, it was 42 years. Regarding education, 9 (43%) of the men have between 6 and 10 years of schooling, while 11 (52%) of the women have between 11 and 15 years of schooling. Concerning occupation, 9 (43%) of the women are involved in agriculture, and 5 (24%) are engaged in household activities. On the other hand, 5 (24%) of the men are involved in agriculture, and 5 (24%) are engaged in fishing activities (Table 1).

## Study design

This participatory research seeks to establish horizontal power relationships between the community and the researchers, to evaluate the context of SBE and validate the most pressing health issues. In this collaborative framework, agenda-setting, problem identification, and methodology development are co-created by both parties. To achieve this, the study was structured into three phases (Fig 2).

The study was divided into three main steps. Step 1, called the Community Baseline Assessment, involved administering questionnaires to collect sociodemographic data, conducting in-depth individual interviews to explore participants' perceptions and experiences, leading focus groups to collectively discuss health issues, and observing the community's interactions and activities to better understand the cultural context and practices. Additionally, a preliminary analysis of the qualitative data collected was performed.

Step 2, the Community Assembly, focused on presenting the health issues identified during the analysis of the first step, followed by seeking consensus on the most pressing health problems, discussing viable solutions, and prioritizing these issues. During this step, qualitative data were also collected through participant observation and researchers' field notes. This phase was guided by the principles of Paulo Freire [31,32], who views education as a practice of freedom, to inform the methodology in identifying the community's primary health challenges, prioritizing the issues, and developing effective strategies to address them. Horizontal relationships were strengthened during this process as participants actively engaged with the preliminary findings, validating results, contributing new insights, and refining the data by removing unnecessary interpretations.

Step 3, the Final Data Analysis, involved method triangulation to ensure the validity and reliability of the data collected in the previous steps. In this phase, a thematic analysis of the qualitative data was conducted, including in-depth interviews, focus group discussions, participant observation, and researchers' field notes. This analysis culminated in the preparation of a report detailing the main problems faced by the community and the strategies and solutions to overcome them.

## Research team and reflexivity

The study team consisted of thirteen members, including biomedical professionals, biologists, pharmacists, nurses, and educators, with seven women and six men, all experienced in qualitative research. The team included five PhD-qualified qualitative researchers Felipe Leão Gomes Murta (FLGM), Wuelton Marcelo Monteiro (WMM), Vinícius Azevedo Machado (VAM),

**Table 1. Sociodemographic characteristics of participants.**

| Participants | Total, n = 42 | |
|---|---|---|
| **Characteristics** | **Female**, n = 21[a] | **Male**, n = 21[a] |
| **Median age (years)** | 30 (22, 39) | 42 (21, 51) |
| **Marital status** | | |
| Single | 8 (38%) | 12 (57%) |
| Married | 11 (52%) | 8 (38%) |
| Widower | 0 (0%) | 1 (4.8%) |
| Other | 2 (9.5%) | 0 (0%) |
| **Education level (years)** | | |
| 1–5 | 2 (9.5%) | 2 (9.5%) |
| 6–10 | 6 (29%) | 9 (43%) |
| 11–15 | 11 (52%) | 7 (33%) |
| >15 | 2 (9.5%) | 3 (14%) |
| **Occupation** | | |
| Housewife | 5 (24%) | 0 (0%) |
| Farmer | 9 (43%) | 5 (24%) |
| Fisher | 0 (0%) | 5 (24%) |
| Teacher | 2 (9.5%) | 2 (9.5%) |
| Other | 5 (24%) | 9 (43%) |
| **Source of income** | | |
| National Institute of Social Security | 3 (14%) | 3 (14%) |
| Self-employed | 8 (38%) | 8 (38%) |
| Public worker | 7 (33%) | 8 (38%) |
| Other source | 0 (0%) | 2 (9.5%) |
| **Family income** | | |
| [b] Up to R$ 500,00 | 2 (9.5%) | 1 (4.8%) |
| [b] Up to R$ 1.100,00 | 12 (57%) | 9 (43%) |
| [b] Up to R$ 2.200,00 | 6 (29%) | 6 (29%) |
| [b] Up to de R$ 3.300,00 | 1 (4.8%) | 4 (19%) |
| [b] More than R$ 3.300,00 | 0 (0%) | 1 (4.8%) |
| **Automobiles** | | |
| Motorcycle | 7 (33%) | 8 (38%) |
| Bicycle | 2 (9.5%) | 1 (4.8%) |
| Rowing canoe | 2 (9.5%) | 5 (24%) |
| Motorized canoe | 21 (100%) | 19 (90%) |

[a] Median (IQR); n (%).

[b] Up to R$500.00 (approximately up to US$92.59); Up to R$1,100.00 (approximately up to US$203.70); Up to R$2,200.00 (approximately up to US$407.41); Up to R$3,300.00 (approximately up to US$611.11); More than R$3,300.00 (more than US$611.11).

Jacqueline Almeida Gomes Sachett (JAGS), Altair Seabra de Farias (ASF) one PhD candidate Alícia Patrine Cacau dos Santos (APCS), four master's degree students Evellyn Antonieta Rondon Tomé da Silva (EARTS), Hiran Sátiro Souza da Gama (HSSG), Jady Shayenne Mota Cordeiro (JSMC), Rafaela Nunes Dávila (RND), and three fieldwork researchers Ana Paula Silva Oliveira (APO), JAA (Jéssica Albuquerque Araújo), Hélio Afonso Amazonas Júnior (HAAJ). The study team had no prior relationships with the participants and made efforts to prevent their perspectives from directly influencing data collection and analysis.

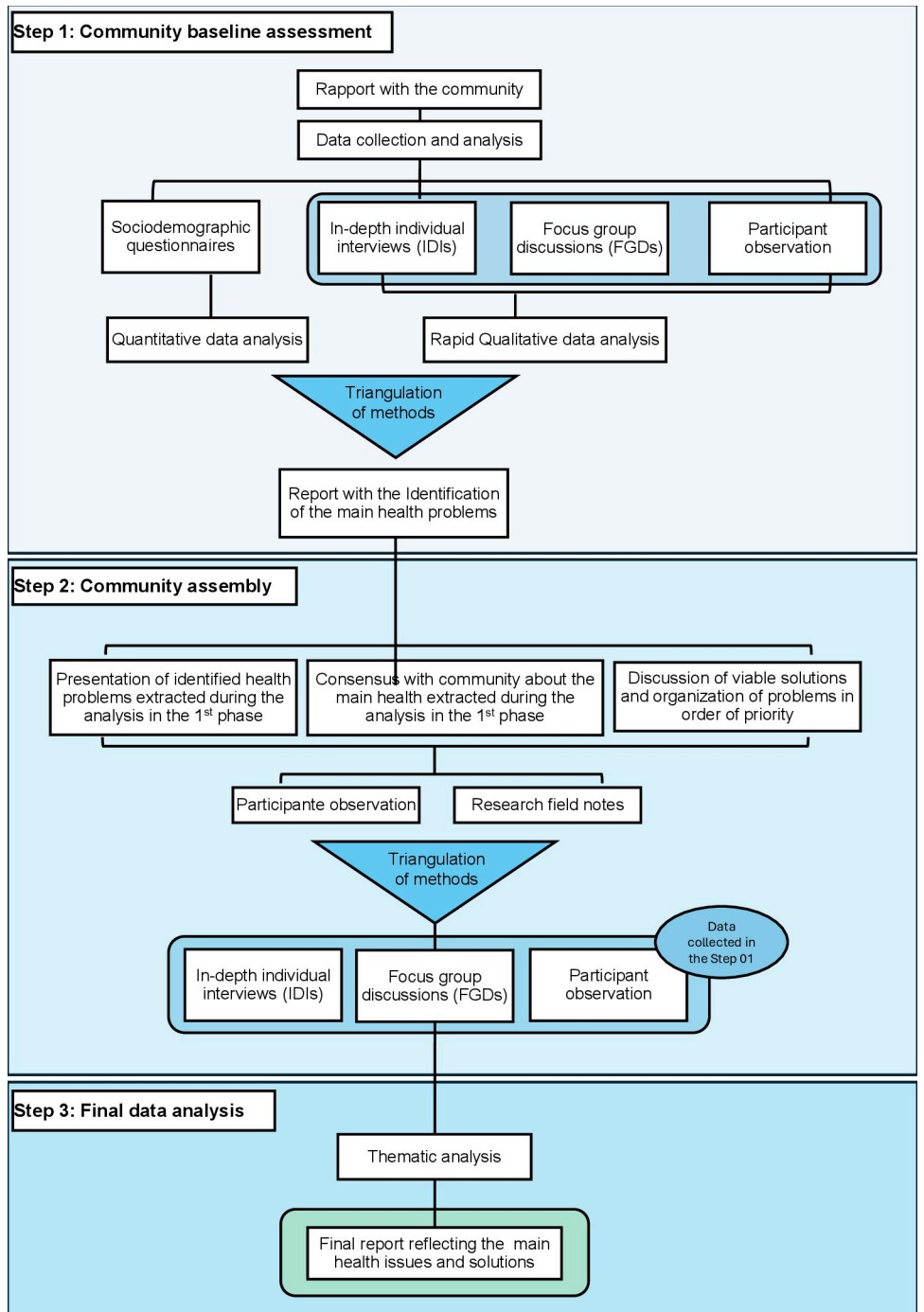

**Fig 2. Flowchart of the study design and data collection.**

## Sampling approach

The study employed a multi-method sampling strategy tailored to each research method to ensure comprehensive data collection and representativeness.

In the cross-sectional survey, a total of 42 participants were recruited using convenience sampling, prioritizing individuals who could provide valuable insights into the research topic.

Convenience sampling was chosen for the cross-sectional survey due to the logistical challenges of conducting research in remote riverine communities. This method enabled the recruitment of participants who were readily available and willing to contribute, ensuring the inclusion of diverse perspectives while prioritizing feasibility. Despite its limitations, such as the potential exclusion of more marginalized groups, this approach was complemented by purposive sampling in qualitative methods, which allowed for deeper exploration of varied experiences and ensured robust data collection. All participants were aged 12 years or older, with informed consent obtained from adults and informed assent from minors, along with consent from their legal guardians.

In the qualitative approach, 32 participants were engaged through purposive sampling, which included 15 individual interviews (IDIs) and three focus group discussions (FGDs). Of the 15 participants in the IDIs, 8 were women aged between 20 and 70 years, and 7 were men aged between 32 and 69 years. The FGDs were conducted in three categories: women's groups (5 participants), men's groups (5 participants), and teenagers' groups (7 participants). The principle of theoretical saturation [33] guided data collection, ensuring patterns emerged clearly across interviews and discussions. Each session was designed to explore health-related perceptions and experiences in the community.

For the community assembly, all 42 participants from the initial phase were invited to participate, but only 20 attended. This assembly allowed the community to collectively prioritize health problems and propose solutions, ensuring their active involvement in decision-making processes. In the participant observation, eight observations were conducted to understand the community's daily activities, interactions, and cultural practices.

### Step 1- Community baseline assessment

**Quantitative and qualitative data collection and analysis.**   The objective of this phase was establishing rapport with the community and collect data. Thus, the first two visits, conducted by ACPS between August/2021 and January/2022, were intended to introduce the project to local leaders, obtain consent, and familiarize the team with the community members. During the initial visit, the research team focused exclusively on rapport building, presenting the project objectives, and emphasizing the benefits of community participation. This included discussing how the study could help identify and prioritize health problems and potential solutions in collaboration with the community. The third visit took place in June/2022 focusing on data collection and involving team members APCS, EARTS, and HSSG. The final visit occurred in September/2022 for the community assembly, attended by ACPS and VAM.

To establish a baseline understanding of the community, structured questionnaires were administered to collect socio-demographic data, as well as previous knowledge and experience regarding SBE (S1 and S2 Files). The quantitative data were then entered into RedCap 13.1.3 (Research Electronic Data Capture) and exported to R software version 4.1. Descriptive statistical analysis was performed on the quantitative data, with all tables generated using R software version 4.1 and RStudio, utilizing the tidyverse, gt gtsummary, and expss packages.

For the qualitative approach, face-to-face In-Depth Interviews (IDIs) and focus group discussions (FGDs) were conducted by APCS, EARTS, and HSSG. A semi-structured interview guide (S3 File) with open-ended questions was developed by the research team, who have expertise in qualitative studies and was validated with a smaller sample of volunteers. This interview guide was developed to elicit detailed insights into participants' perceptions, experiences, and practices related to health issues within the community. The open-ended questions were structured around key themes identified as relevant by the research team, such as access to healthcare, common illnesses, prevention practices, and community perspectives on SBE.

The IDIs were held in participants' homes, ensuring a comfortable and quiet environment only with the interviewer, the observer to take field notes and the participant. Each interview lasted approximately 60 minutes. With participants' consent, the interviews were recorded in audio and fully transcribed for subsequent analysis. The number of interviews was determined by the principle of theoretical saturation where IDIs are carried out until a clear pattern appears and subsequent groups do not produce new information.

The FGDs were organized into three distinct categories: women, men, and adolescents, each consisting of five to seven participants. These discussions followed a structured protocol designed to encourage collective dialogue on key themes identified during the individual interviews. Experienced researchers moderated the FGDs to create an atmosphere where all participants felt comfortable sharing their opinions. Each FGDs session lasted approximately 80 minutes, and, like the IDIs, they were recorded in audio and transcribed for analysis.

Qualitative data analysis and interpretation occurred simultaneously in Step 1 of the study through rapid qualitative analysis, utilizing debriefings [34]. This approach was selected for its ability to provide immediate insights during data collection, allowing for real-time adjustments. After each IDI and FGD, audio recordings and field observations were reviewed and discussed by APCS, EARTS, and HSSG. The debriefing sessions offered immediate insights into the data content and facilitated necessary adjustments for deeper exploration during data collection. Data analysis involved discussions among field researchers, reviews of interviews, audio recordings, and field observations. Following these discussions, inductively extracted data were organized in a Word document, which later generated a report on the key health issues faced by the community.

**Preparation of the step 2.**   At the end of Step 1 of qualitative and quantitative data collection and analysis, a report was prepared (S4 File). This report contained a detailed analysis of the main health problems identified in the community, their knowledge about SBE, and other relevant information, such as socio-demographic data, preventive behaviors, and barriers to accessing healthcare. This information was crucial in supporting Step 2 of the study, which was conducted through a community assembly. During this assembly, the report's findings were presented and discussed with the community to validate the findings, establish priorities, and develop joint strategies to address the identified health challenges.

## Step 2 –Community assembly

**Qualitative data collection and analysis.**   The objective of this step was to build consensus with the community regarding the health problems identified in Step 1, ensuring these issues were relevant and prioritized by the community. The assembly was conducted using Paulo Freire's culture circles, a participatory approach that fosters learning by addressing shared experiences, valuing local culture, and promoting a horizontal educator-learner relationship [31].

The process followed three stages: generative theme, where existing knowledge was explored; thematization, analyzing this knowledge for social meaning; and problematization, transcending initial perspectives to deepen understanding of the lived context. These stages emphasized dialogue and collective problem-solving, ensuring an open and horizontal discussion where participants shared their experiences and perspectives.

The assembly began with a presentation of the health problems identified in Step 1, followed by discussions to refine the list, include additional suggestions, and address critiques. Participants worked with researchers to prioritize the issues and explore viable solutions. After reaching a consensus, APCS prepared a detailed report (S5 File), summarizing the health problems deemed important by the community and outlining proposed solutions for each issue, based on field notes and observations.

### Step 3 – Final data analysis

In Step 3, we conducted a final analysis of the data to deepen the understanding of health issues within the community, using a methodological triangulation approach. Initially, all qualitative data collected in Step 1, including IDIs and FGDs, were fully transcribed without personal identifiers and entered into MAXQDA 20 software for detailed analysis.

Methodological triangulation involved integrating the data collected in Step 1 with the report produced by APCS. This combination allowed for the verification and enrichment of the initial findings in light of the collective discussions held in the community, validating the emerging themes and ensuring that both individual and collective perceptions were considered.

Braun and Clarke's thematic analysis [35,36] was conducted using MAXQDA 20 software, through deductive coding. The data were coded by APCS, and the preliminary codebook was reviewed by FLGM. Subsequently, the coded segments were grouped into categories and reviewed (S6 File). After constructing the categories and discussions, the themes and sub-themes were developed. The final analysis generated a detailed description of the health problems faced by the community and the solutions identified by the community members in both stages.

## Results

### Identification of the main health problems

During the community assembly, the study participants discussed the issues identified by the researchers in Step 1 and ranked their importance within the community in order of priority (Table 2). The participants agreed that the problems identified by the researchers were part of their daily lives, and were important, and they also suggested solutions for each issue.

Regarding the first health issue listed, accidents caused by venomous animals, the descriptive statistical analysis revealed that participants had contact with various animals, including snakes, stingrays, spiders, scorpions, and tucandeira ants (Table 3). These findings were further explored during the qualitative data collection.

To understand the issues related to SBE, 36 participants (92%) stated that they had encountered a snake at some point in their lives. Additionally, 41 participants (98%) considered it a serious health problem for the population, and all 42 participants (100%) believed that hospital treatment was necessary. However, 26 participants (62%) believed that using a tourniquet could assist in treatment.

Regarding the practice of walking barefoot in the community, 21 individuals (50%) indicated that it was a common practice, especially near their homes. However, participants also

**Table 2. Order of Priority of Issues Defined by the Riverine Community.**

| Order of priority | Highlighted issues |
|:---:|:---|
| 1st | Accidents involving venomous animals |
| 2nd | Women's health |
| 3rd | Viral illnesses |
| 4th | Accidents with sharps and objects |
| 5th | Fractures and sprains |
| 6th | Drownings |
| 7th | Domestic accidents |
| 8th | Sexually transmitted infections (STIs) |
| 9th | Seizures |

**Table 3. Study participants' exposure to venomous animals.**

| Experience with venomous animals | n = 42 |
|---|---|
| **Affected participants** | 10 (23,8%) |
| **Animal** | |
| **Snakes** | 2 (4,7%) |
| **Stingray** | 2 (4,7%) |
| **Spider** | 2 (4,7%) |
| **Scorpions** | 2 (4,7%) |
| **Tucandeira ants** | 2 (4,7%) |

reported that they take specific precautions, such as wearing protective gear, during activities perceived as high-risk, like farming or fishing. Engaging in activities close to vegetation was reported by 28 participants (68%).

As a preventive measure, 33 participants (79%) reported using high boots, while only 14 participants (34%) mentioned using thick gloves when handling dry leaves and piles of trash. Additionally, 39 participants (93%) stated that they seek medical help when necessary. These quantitative findings highlight variations in attitudes and practices shaped by occupation, gender, and cultural beliefs. Among those who reported using high boots, 81% were men, reflecting the nature of their tasks, which often involve greater exposure of the lower limbs to risks. In contrast, 87% of those who mentioned using thick gloves were women, likely due to their frequent involvement in vegetable cultivation, a task that exposes their hands to potential hazards, justifying the higher use of gloves. Furthermore, many women expressed fear of snakes and reliance on men to handle encounters, highlighting a gender dynamic in risk perception and response.

The quantitative data from the study reveal that the community has a significant history of SBE, with 26 participants (61%) reporting past experiences, primarily involving family members. These incidents, usually affecting the legs and feet, resulted in common symptoms including swelling, redness, vomiting, chills, fever, and nosebleeds. Most of the snakes involved were from the *Bothrops atrox* and *Micrurus* species. Among the SBE victims reported by the participants, 16 fully recovered, while 10 continued to experience lingering effects.

**Thematic analysis of community health problems and solutions.**    In total, 32 people participated in the interviews conducted in Step 1, which included 15 IDIs and 3 FGDs. The FGDs were divided into Women's groups (5 participants), Men's groups (5 participants), and Teenagers' groups (7 participants). Meanwhile, 20 residents participated in the community assembly, including 12 men and 8 women; all these individuals also participated in the data collection in Step 1. The themes outlined below were derived from the IDIs, FGDs, and community assembly following data analysis, with the findings summarized in Fig 3.

## Theme 1: SBEs in the community

**Sub-theme 1a: Experiences with SBE.**    Most participants perceive SBE as a severe health problem that can lead to death if not treated correctly. Regarding experiences with snakes in the community, several participants reported encountering venomous or non-venomous snakes multiple times throughout their lives. These encounters often occurred during daily activities such as agriculture, fishing, inside their homes, in the yard, on the way to school, and at church.

- **Theme 1: SBEs in the community**
  The first theme covers mostly experiences with snakes, the preventive measures adopted by the riverine, and self-care in this cases. Were also extracted four sub-themes from the data:
    - Sub-theme 1a: Experiences with SBEs
    - Sub-theme 1b: Preventions measures for SBEs
    - Sub-theme 1c: Beliefs about SBEs
    - Sub-theme 1d: Knowledge about treatmente and self care.

- **Theme 2: Common health issues in the community**
  The second theme addresses other problems health problems faced by the community such as: women's health, viral illnesses, accidents with sharp objects, fractures and sprains, drownings, domestic accidents, STIs and seizures.

- **Theme 3: Community defining solutions**
  The third theme refers to the main strategies and solutions identified by community members to address the health problems encountered. Were also extracted two sub-themes from the data:

    - Sub-theme 3a: First aid training and emergency communication
    - Sub-theme 3b: Access to healthcare and transportation

**Fig 3. Themes and sub-themes identified during thematic analysis.**

*"Just the other day, on top of my bathroom, I reached for the tap, and then I felt something curled up here. It was a snake. I screamed so loudly that my voice almost didn't return to normal."* (Participant 29, woman, farmer).

The riverine people have extensive experience with accidents, as a large portion of them have a family member who has suffered an accident and required hospital assistance in the past, as described below:

*"This happened to my mother [. . .] the snake that bit her was small. I think it was about this size, not a big snake. She had gone to pick some oranges, and she felt when it bit her, but she didn't know exactly what it was, you know, because it was kind of dense"* (Participant 1, woman, farmer).

Participants reported that, beyond SBE, accidents involving other venomous animals are routine, with common species including ants, caterpillars, bees, spiders, scorpions, centipedes, stingrays, and spiny fish during fishing.

*"When I was in Bom Futuro [a neighboring riverine community], an ant bit me. Now, centipedes bite me too, but sometimes it affects some people, and not others "*(Participant 14, male, farmer).

**Sub-theme 1b: Prevention measures for SBE.**   Participants noted that SBE is rare, primarily because they usually wear protective gear such as boots, long pants, long-sleeved shirts,

and carry machetes. They also approach all their activities with caution and attentiveness to avoid accidents.

*"We need to be properly equipped with boots, long pants, and long-sleeved shirts. Even if your foot is covered, it's still dangerous because a jararaca doesn't just bite; it can leap quite far"* (Participant 3, female, farmer).

One commonly adopted prevention measure is to kill the snake whenever it is encountered. Some participants acknowledged that environmental regulations discourage killing snakes. However, they mentioned that in real-life situations, their perspective differs because snakes are seen as dangerous and unpredictable. Their initial thought is often, "better to kill it before it kills me." Despite their extensive experience with handling snakebite victims, men, in particular, reported not feeling fear of the animal, as snakes are a constant presence in their daily lives. Conversely, women expressed fear of the animal and often rely on a male figure to resolve the situation, typically by killing the snake.

*"It might be prohibited, but if I see a snake that I consider dangerous and that could harm me or others, I kill it before that happens. [. . .] I'm not afraid because I know how to act and defend myself. [. . .] Usually, the snake, it doesn't come after you to bite; it only bites when, sometimes, you step too close to it or step on it. If you step on it, you're sure to get bitten"* (Participant 14, male, farmer).

**Sub-theme 1c: Beliefs about SBE.**   There are several beliefs surrounding SBE, including the idea that children born to mothers who were bitten by snakes during pregnancy are immune to venom and are born cured. In such cases, people who have been bitten by snakes seek out these cured individuals to receive a kind of antivenom by having them spit on the wound, as described in the following excerpts:

*"They say that when a woman is pregnant and gets bitten by a snake, the child comes out cured, free from all creatures. In my nephew's case, I don't know if she was bitten by a snake or what, but he doesn't feel anything"* (Participant 10, female, general services assistant).

Additionally, there is a belief that one should not leave a snake alive because it will wait in the same place on the seventh day after the encounter to strike again, as described in the following excerpts:

*"According to human law, you're not supposed to kill them, but. . . I don't leave them alive. [. . .] Sometimes, they don't run away; they stay there, ready to attack"* (Participant 4, male, fisherman).

Participants also believe that snakes can be enchanted beings that sing after biting. Furthermore, there is a belief in the existence of a large snake in the Solimões River that should be respected and feared because it tends to take people into the depths of the river.

*"There's a type of snake, a greenish one [Corallus batesii], and it sings like a parrot when it bites. My mom says that when it bites someone and sings, the person won't survive; they'll die"* (Participant 10, female, general services assistant).

**Sub-theme 1d: Knowledge about treatment and self-care.** Participants understand that effective and proper treatment is administered in a hospital using antivenom. However, they lack a clear understanding of how the treatment works.

*"I know they inject antivenom into the person, but I don't know about the rest of the treatment, whether they prescribe medication or not"* (Participant 11, female, farmer).

Participants also described several precautions, first aid measures, and recovery procedures to follow after a snakebite. These actions involve making incisions, applying tourniquets, and adhering to a diet free of fatty foods.

*"When a snake bites you, it starts numbing everything. [. . .] That's why we tie the cloth here, so the blood doesn't spread that way, because when it bites, the venom goes into the bloodstream immediately. So, you have to tie it very tightly"* (Participant 4, male, fisherman).

**Theme 2: Common health issues in the community.** The riverine community faces various health problems, including women's health issues, viral illnesses, accidents involving sharp objects, fractures and sprains, drownings, domestic accidents, STIs, and seizures (Table 2).

Women highlighted delays in obtaining cytological exams, mammograms, and access to contraceptives as significant health concerns. Intestinal viruses in children are another worry, especially during the dry season when clean rainwater is scarce, forcing residents to use river or communal well water. Accidents involving sharp objects are common during agricultural and fishing activities, often resulting in deep cuts. These accidents also occur when clearing areas around homes due to hazards like nails, glass bottles, needles, and coins. Fishing accidents involving hooks are particularly dangerous, especially when fishermen are adrift in rivers. Fractures and sprains, primarily occurring during fruit collection, are frequent, with participants often unsure how to treat them.

Drownings typically involve children during leisure activities or when accompanying mothers who are washing clothes by the river. These incidents also occur during fishing trips due to storms, landslides, or strong currents. Domestic accidents, especially those involving cuts and burns from hot oil, boiling water, caramelized sugar, and kerosene, are a daily occurrence. Seizures were also highlighted as a priority health issue in the community (as shown in Table 4).

## Theme 3: Community defining solutions

The participants of the study mapped out a series of strategies and solutions (Table 5) for the problems they face, according to their perspectives. Among the most frequently mentioned strategies were: receiving first aid training, installing a satellite phone for effective communication with SAMU (Mobile Emergency Medical Service), as well as actions dependent on local administration, such as constructing a basic health unit, improving the access road to the community, hiring healthcare professionals, and providing better access to medications.

## Sub-theme 3a: First aid training and emergency communication

The participants expressed a strong desire for information and training on first aid, particularly for accidents, drownings, seizures, burns, and other emergencies. They believe that having knowledge about first aid would empower them to respond appropriately in critical situations, especially when immediate help is not readily available.

**Table 4. Health concerns reported by participants from the community.**

| Problems | Quotes |
|---|---|
| Women's health | "To get an injection [contraceptive], sometimes we have to go to Tabatinga. Sometimes I send my brother-in-law because he has experience giving injections" (Participant 31, female, farmer). |
| Viral illnesses | "The most common health problems here are fever, flu, diarrhea, and recently, stomachaches" (Participant 8, female, farmer). |
| Accidents with sharp objects | "When using a weed trimmer, you have to wear goggles because sometimes a piece of wood flies into your face" (Participant 29, female, farmer). |
| Fractures and sprains | "I was chasing after parrots with some guys whose names I didn't know, and I hit my head on a post, broke it. . . blood started flowing" (Participant 21, male, student). |
| Drownings | "I almost drowned once. . . I was crossing the river, and it was agitated. . . I was rowing with a paddle because we didn't have a motor yet, and then the canoe flipped, and I went deep, but I had an ice chest in the canoe for the fish, so I went on top of it and screamed in the middle of the river" (Participant 18, male, farmer). |
| Domestic accidents | "I was frying some dumplings, and when I went to flip them, there was too much oil. When I took them out, it splashed on my arm, leaving a mark here on my arms" (Participant 25, female, student). |
| STIs | "I always talk with my kids about STIs because I have a son who is single, so I always tell him: 'My son, you guys have to be careful, take precautions; if you're going to do something, use protection because nobody knows about the other" (Participant 3, female, farmer). |
| Seizures | "Epilepsy is also a concern. . . I have a nephew who is epileptic, and sometimes we must be careful. It would be good if we could learn about this too" (Participant 19, male, household). |

*"Teach first aid because, as it is very difficult for us to go to the city, if we could help right here. . . If an accident happens, we don't know [how to respond]"* (Participant 04, male, fisherman).

**Table 5. Solutions identified by community assembly.**

| Problems | Solutions identified by community members |
|---|---|
| Accidents by venomous animals, especially SBE | • Annual monitoring of venomous animals and accidents.<br>• The city government should improve the access road from the community to Tabatinga.<br>• The city government should build a basic health unit in the community.<br>• Hire a nursing technician with first aid expertise for the community.<br>• Acquire a satellite phone (rural phone) for the community. |
| Women's health | • A trained person to administer contraceptive injections to women in the community. |
| Illness viruses | • The community should receive first aid training.<br>• Better access to medications.<br>• The community should receive guidelines for potable water treatment. |
| Accidents with sharps and objects | • The riverine community must perform activities with greater attention.<br>• The community should receive first aid training. |
| Fractures and sprains | • The community should receive first aid training.<br>• The community should have access to a form of transportation for reaching health facilities.<br>• Acquire a satellite phone (rural phone) for the community. |
| Drownings | • The community should receive first aid training. |
| Domestic accidents | • The community should receive first aid training. |
| STIs | • Distribution of condoms.<br>• The community should receive guidance about condoms and sexual health. |
| Seizures | • The community should receive first aid training. |

*"Teaching this kind of first aid would be essential because sometimes when someone gets hurt, the nervousness prevents us from thinking clearly. So, it would be good"* (Participant 11, female, farmer).

They also reported that the phone signal is poor in the community, sometimes requiring them to find a specific location where the signal works to make a call and request assistance from SAMU (Mobile Emergency Medical Service) in Tabatinga. They believe that having a satellite phone would make communication more reliable.

*"If there were a satellite phone, we wouldn't have this issue of searching for a specific place in the community to see if it works when making a call. It would be available; it would be easier than finding a place. It would stay connected, making communication easier for us"* (Participant 9, male, farmer).

### Sub-theme 3b: Access to healthcare and transportation

The participants emphasized the importance of constructing a health unit closer to their community to serve all nearby riverine communities. This would alleviate the difficulties they currently face in accessing healthcare. They also stressed the need for specialized healthcare professionals, particularly for family planning and the administration of contraceptives to women in the community, who reported challenges in this regard.

*"So, if there were a health center, a nurse would help a lot, but there has to be a base, a structure to make it work"* (Participant 5, male, fisherman).

*"If there were a health center, it would be a big help, and we've been requesting a health post for a long time now"* (Participant 6, male, farmer).

Transportation was another major concern. The community reported the need for a boat, especially during the rainy season, and a kind of motorcycle for use during the dry season called a motorcar, a vehicle that is part car and part motorcycle. They also noted that the boat should be available to the community health agent. The following passages illustrate this scenario:

*"I see that our situation here, what's lacking is a strong boat. If we had at least a 40-horsepower boat with a barge, because there are cases here where sometimes a person arrives at the UPA or the Hospital [without transportation]"* (Participant 13, male, elementary school teacher).

The participants also mentioned that improved access to basic medications would be a significant improvement since there is no pharmacy or drugstore in the community. They reported difficulties obtaining analgesics, antipyretics, antiparasitic medications, and materials for wound care, such as bandages, alcohol, adhesive tape, and ointments. Additionally, they expressed frustration with obtaining medications at the health post, as prescriptions are required for everything, leading to medications expiring on the shelves.

*"I would say that if you could provide guidance on medications, it would help a lot because, at the health centers, they are not even providing medications to the health agents anymore, as they used to with some types of medications"* (Participant 08, female, farmer).

## Discussion

### The place of SBE in the community

SBE primarily occurs within the community, with most reported cases dating back over 10 years. This decline in incidents may be attributed to the increased use of protective equipment during agricultural activities and the heightened awareness reported by community members. One factor emphasized by participants was the greater accessibility to protective equipment, such as boots and gloves, whose prices have reportedly decreased over the past five years. This reduction in cost allowed more families to acquire and use these items in high-risk activities, especially in agriculture. Participants also demonstrate a strong understanding of the severity of SBE accidents and the importance of prompt medical treatment, which likely contributes to the rarity of recent cases. Studies emphasize the use of boots during daily activities to reduce SBE and highlight the role of educational initiatives in raising awareness about prevention and first aid [37,38].

Contact with snakes is frequent, with a significant number of participants reporting at least one encounter with a snake in their lifetime. However, actual incidents are rare, as participants often kill snakes upon encounter to protect themselves. The frequent contact with snakes is attributed to various daily tasks, such as inserting hands into tree trunks in search of bait for fishing, walking to school or church, or during agricultural activities like planting. These findings are consistent with studies conducted in the Brazilian Amazon, which reveal that accidents can occur during routine activities or around the home, places typically considered safe, where vigilance may decrease, leading to accidents [1–3]. Also, climate change and environmental degradation, such as deforestation, are expanding the geographic range and activity of venomous snake species, increasing human-snake encounters and the risk of envenomation. As highlighted by WHO [39] and supported by recent predictive models [40] rising temperatures and habitat changes exacerbate these risks, particularly in tropical regions where biodiversity and rural livelihoods are most vulnerable. These environmental changes also heighten exposure to other health issues, including infectious diseases linked to ecosystem disruptions.

SBE is intertwined with a range of other common issues that are often considered higher priorities. Participants noted that SBE is not the only significant concern. The inclusion of other common health issues, such as women's health, viral illnesses, and accidents, provides a broader framework for understanding and addressing healthcare challenges in the community. Addressing these interconnected health problems creates opportunities to strengthen healthcare systems and infrastructure, which, in turn, benefit SBE management.

In addition to venomous animal-related issues, participants reported other health problems that are often overlooked by researchers, such as women's health, sexually transmitted diseases, work accidents, and domestic accidents. Health surveys conducted in riverine areas have revealed a series of health issues, including a high prevalence of chronic diseases such as hypertension and diabetes, particularly among the elderly [41,42]. Besides that, a significant number of women are infected with HPV, a risk factor for cervical cancer. Brazil is the third country with the most incidences of this disease among the female population in the world and it is the fourth leading cause of death among women, resulting in more than 6,000 deaths annually [43,44]. Additionally, studies on the sexual and reproductive health of women in these areas are still scarce [45].

### The effects of invisibility in the access to primary health care services

The Unified Health System (SUS) aims to ensure universality, comprehensiveness, and equity, particularly through initiatives like the Riverine Family Health Teams (eSFR) and Basic Fluvial

Health Units (UBSF), which operate via river access using boats equipped with healthcare facilities [46,47]. However, many communities, such as Limeira, still face challenges in accessing healthcare services due to financial and logistical barriers. A Ministry of Health workshop in 2022 highlighted the extended travel times required for healthcare professionals, sometimes taking 24 to 48 hours to reach remote communities, highlighting the need for more decentralized primary care [48]. Furthermore, the limited availability of fluvial ambulances and the communication difficulties with emergency care services further complicate healthcare delivery in these regions [49].

The need for urgent and emergency care is a reality in riverine areas, which face numerous barriers to effective healthcare delivery. In particular, tropical diseases and accidents involving venomous animals must be taken into account, as these incidents are common and can lead to severe complications or death if specialized care is not promptly administered. Studies indicate that a significant number of individuals die without receiving appropriate treatment in cases of SBE [4,50], which can also occur in accidents involving scorpions, especially in children, anaphylactic reactions from bee stings, severe hemorrhages caused by caterpillars, and other similar incidents. Therefore, decentralizing specialized care to primary health units is a solution worth considering, particularly in situations requiring rapid intervention. A recent study in Brazil proposed the decentralization of antivenom distribution to riverine and indigenous communities in the Brazilian Amazon, aiming to reduce the burden of SBE in the region [51].

The barriers mentioned above, coupled with existing inequalities, underscore the inequities in access to the SUS, compromising the provision of both primary and specialized care, particularly for marginalized populations. The state of Amazonas has one of the lowest percentages of coverage in primary care, second only to Pará [52]. Studies conducted near the state capital, Manaus, confirm this reality and reveal a significant number of individuals who have never had contact with a doctor or dentist in their lifetime, a situation closely linked to social and economic inequalities [53]. Additionally, services offered in riverine areas continue to be planned hierarchically, with their organization and execution based on urban realities [17].

This study has some limitations that should be considered. The findings reflect the specific realities of the Limeira community and may not be generalizable to other riverine populations in the Amazon, given the unique cultural and geographical characteristics of each community. Moreover, while efforts were made to minimize biases, factors such as age and gender dynamics in the focus groups may have influenced participants' responses, particularly when describing the behaviors of others. Furthermore, challenges encountered during data collection, such as logistical constraints and variations in participant availability, may have affected the depth and scope of the qualitative data. These limitations underscore the need for further studies involving diverse riverine communities to develop a more comprehensive understanding of health challenges and interventions in the Brazilian Amazon. Another important consideration is the potential for social desirability bias, given the researchers' stated interest in SBE. While participants reported SBEs as rare events in the past 10 years, their classification as a top priority may reflect the historical perception of the problem or the emotional impact associated with severe cases. This discrepancy between recent frequency and perceived severity highlights the need to interpret the data within the cultural and social context of the community. The decrease in the number of participants from the survey (n = 42) to the community assemby (n = 20) is a limitation that can be partly attributed to logistical factors related to monthly travel, especially at the end of the month. During this period, people receive their salaries, pensions, social benefits, and have better sales of fish, vegetables, and fruits. This makes attending the meeting more difficult, particularly around the end of the month or during festive events, when people may have other personal commitments, they consider more important. To mitigate this, we suggest considering scheduling community meetings at times that

avoid dates close to the end of the month or during local festivities. Additionally, it would be helpful to make community events more accessible in terms of timing and format, such as offering sessions at different times or shorter events. These approaches could help improve participation and engagement in future meetings.

## Conclusion

This study provides valuable insights into the health conditions of a riverine community in the Brazilian Amazon, emphasizing the significant barriers to healthcare access, including logistical challenges, financial limitations, and communication issues. The findings highlight the importance of developing health strategies that are specifically tailored to the community's cultural, historical, and ecological contexts. In particular, the study underscores the need for improved healthcare access and the relative importance of addressing snakebite envenoming (SBE) alongside other health concerns that are often neglected in such populations. These results contribute to a deeper understanding of the unique health challenges faced by riverine communities and offer evidence-based knowledge for future healthcare interventions. The findings suggest that tailored, context-sensitive health strategies are essential for effectively addressing the health needs of these vulnerable populations. A critical evaluation of the methods used in this study, particularly the participatory approach, will be discussed in a separate manuscript.

## Supporting information

**S1 File. Socio-demographic questionnaire.**
(PDF)

**S2 File. Previous knowledge and experience about SBE.**
(PDF)

**S3 File. Semi-structured interview guide.**
(PDF)

**S4 File. Report prepared at the end of step 1.**
(PDF)

**S5 File. Report from the community assembly.**
(PDF)

**S6 File. Codebook from qualitative analysis.**
(PDF)

**S7 File. COREQ. Consolidated criteria for reporting qualitative studies.**
(PDF)

## Acknowledgments

We thank all the riverine dwellers, because their participation was crucial for the development of this study.

## Author Contributions

**Conceptualization:** Alícia Patrine Cacau Santos, Vinícius Azevedo Machado, Wuelton Marcelo Monteiro, Felipe Leão Gomes Murta.

**Data curation:** Alícia Patrine Cacau Santos, Vinícius Azevedo Machado, Felipe Leão Gomes Murta.

**Formal analysis:** Alícia Patrine Cacau Santos, Jady Shayenne Mota Cordeiro, Wuelton Marcelo Monteiro, Felipe Leão Gomes Murta.

**Funding acquisition:** Felipe Leão Gomes Murta.

**Investigation:** Alícia Patrine Cacau Santos, Evellyn Antonieta Rondon Tomé Silva, Hiran Sátiro Souza da Gama, Jady Shayenne Mota Cordeiro, Ana Paula Silva Oliveira, Jéssica Albuquerque Araújo, Rafaela Nunes Dávila, Hélio Afonso Amazonas Júnior, Altair Seabra Farias, Jacqueline Almeida Gonçalves Sachett, Vinícius Azevedo Machado, Wuelton Marcelo Monteiro, Felipe Leão Gomes Murta.

**Methodology:** Alícia Patrine Cacau Santos, Evellyn Antonieta Rondon Tomé Silva, Hiran Sátiro Souza da Gama, Jady Shayenne Mota Cordeiro, Ana Paula Silva Oliveira, Jéssica Albuquerque Araújo, Rafaela Nunes Dávila, Hélio Afonso Amazonas Júnior, Altair Seabra Farias, Jacqueline Almeida Gonçalves Sachett, Vinícius Azevedo Machado, Wuelton Marcelo Monteiro, Felipe Leão Gomes Murta.

**Project administration:** Felipe Leão Gomes Murta.

**Supervision:** Felipe Leão Gomes Murta.

**Validation:** Alícia Patrine Cacau Santos, Wuelton Marcelo Monteiro, Felipe Leão Gomes Murta.

**Writing – original draft:** Alícia Patrine Cacau Santos, Evellyn Antonieta Rondon Tomé Silva, Hiran Sátiro Souza da Gama, Jady Shayenne Mota Cordeiro, Ana Paula Silva Oliveira, Jéssica Albuquerque Araújo, Rafaela Nunes Dávila, Hélio Afonso Amazonas Júnior, Altair Seabra Farias, Jacqueline Almeida Gonçalves Sachett, Vinícius Azevedo Machado, Wuelton Marcelo Monteiro, Felipe Leão Gomes Murta.

**Writing – review & editing:** Alícia Patrine Cacau Santos, Evellyn Antonieta Rondon Tomé Silva, Hiran Sátiro Souza da Gama, Jady Shayenne Mota Cordeiro, Ana Paula Silva Oliveira, Jéssica Albuquerque Araújo, Rafaela Nunes Dávila, Hélio Afonso Amazonas Júnior, Altair Seabra Farias, Jacqueline Almeida Gonçalves Sachett, Vinícius Azevedo Machado, Wuelton Marcelo Monteiro, Felipe Leão Gomes Murta.

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
