## [Decision Letter · Decision Letter 0]

27 Nov 2024

PNTD-D-24-01262

Participatory research towards the control of snakebite envenoming and other illnesses in a riverine community of the Western Brazilian Amazon: ‘Turning the invisible into visible’

Dear Dr. Gomes Murta,

Thank you for submitting your manuscript to PLOS Neglected Tropical Diseases. After careful consideration, we feel that it has merit but does not fully meet PLOS Neglected Tropical Diseases's publication criteria as it currently stands. Therefore, we invite you to submit a revised version of the manuscript that addresses the points raised during the review process.

Please submit your revised manuscript within 60 days Jan 26 2025 11:59PM. If you will need more time than this to complete your revisions, please reply to this message or contact the journal office at plosntds@plos.org. Please include the following items when submitting your revised manuscript:

We look forward to receiving your revised manuscript.

Kind regards,

Soumyadeep Bhaumik, MBBS, M.Sc, PhD

Academic Editor

José María Gutiérrez

Section Editor

Shaden Kamhawi

co-Editor-in-Chief

Paul Brindley

co-Editor-in-Chief

**Journal Requirements:**

1) Please upload all main figures as separate Figure files in .tif or .eps format. For more information about how to convert and format your figure files please see our guidelines: 

2) Some material included in your submission may be copyrighted. According to PLOSu2019s copyright policy, authors who use figures or other material (e.g., graphics, clipart, maps) from another author or copyright holder must demonstrate or obtain permission to publish this material under the Creative Commons Attribution 4.0 International (CC BY 4.0) License used by PLOS journals. Please closely review the details of PLOSu2019s copyright requirements here: PLOS Licenses and Copyright. If you need to request permissions from a copyright holder, you may use PLOS's Copyright Content Permission form.

Potential Copyright Issues:

i) Please confirm (a) that you are the photographer of 1, or (b) provide written permission from the photographer to publish the photo(s) under our CC BY 4.0 license.

3) We note that your Data Availability Statement is currently as follows: "All relevant data are within the manuscript and its Supporting Information files.". Please confirm at this time whether or not your submission contains all raw data required to replicate the results of your study. Authors must share the “minimal data set” for their submission. PLOS defines the minimal data set to consist of the data required to replicate all study findings reported in the article, as well as related metadata and methods (https://journals.plos.org/plosone/s/data-availability#loc-minimal-data-set-definition).

4) Please amend your detailed Financial Disclosure statement. This is published with the article. It must therefore be completed in full sentences and contain the exact wording you wish to be published.

1) State the initials, alongside each funding source, of each author to receive each grant. For example: "This work was supported by the National Institutes of Health (####### to AM; ###### to CJ) and the National Science Foundation (###### to AM).".

5)Please ensure that the funders and grant numbers match between the Financial Disclosure field and the Funding Information tab in your submission form. Note that the funders must be provided in the same order in both places as well.

Please indicate by return email the full and correct funding information for your study and confirm the order in which funding contributions should appear. Please be sure to indicate whether the funders played any role in the study design, data collection and analysis, decision to publish, or preparation of the manuscript.

6) Regarding the map in Figure 1: please (a) provide a direct link to the base layer of the map (i.e., the country or region border shape) and ensure this is also included in the figure legend; and (b) provide a link to the terms of use / license information for the base layer image or shapefile.

**Reviewers' Comments:**

Reviewer's Responses to Questions

**Key Review Criteria Required for Acceptance?**

**Methods**

-Are the objectives of the study clearly articulated with a clear testable hypothesis stated?

-Is the study design appropriate to address the stated objectives?

-Is the population clearly described and appropriate for the hypothesis being tested?

-Is the sample size sufficient to ensure adequate power to address the hypothesis being tested?

-Were correct statistical analysis used to support conclusions?

-Are there concerns about ethical or regulatory requirements being met?

Reviewer #1: The objective of this study is clear, aiming to identify and address health issues faced by the Limeira community in the Brazilian Amazon basin through participatory research, particularly focusing on snakebite envenomation (SBE) and other common health problems. However, despite the clarity of the goal, the study does not present a clear testable hypothesis. This raises a question: in such a complex social and health context, should hypotheses be more explicitly defined to guide the research direction, especially when dealing with numerous unknown health challenges? I suggest that the authors provide a broader social context in the 'Introduction' section, rather than only discussing the Amazon region.

Reviewer #2: 240: Purposive sampling is a sound approach to this type of study. Further explanation of questionnaire design and purpose of each question would be helpful. Research design is sound, with appropriate methods used to construct and test the SSI instrument.

270: What type of recordings? Audio? Video?

275: Focus groups typically consists of a dominant speaker who controls the conversation, thus essentially resulting in the equivalent of one SSI and diminishing the sample size.

308 - Was cultural consensus analysis à la Romney, Weller and Batchelder ever calculated from the data? This result could then be compared with the outcome of Freire’s consensus building method results.

376 – Was the ranking task only conducted at the focus group level? It would be interesting to compare group results with those of individuals as a collective as a validity test.

416 – The information on number of interviews and how focus groups were determined should come sooner. While it is not stated, it appears the researchers were interested in comparing gender and age as variable; if so, this should be stated outright. (There is no discussion of differences of variables vis-à-vis FGs anywhere in the paper).

429 – Need a line or two stating that themes are being presented drawn from either just the SSIs, or both the SSIs and FGs.

461 - On page 400, It is presented that 50% of residents report walking barefoot, but on 461 it states that participants said SBE is rare due to people wearing protective gear. This discrepancy may be an artefact of the different data collection methods. Some explanation of this discrepancy, especially the possible reason for it, is warranted.

537- If other common health issues are to be included, they should be directly related to SBE, and the purpose of their inclusion stated – for example, lack of treatment of other health problems correlating with lack of treatment of SBE, etc. (Table 5 makes it seem that there is not a great deal of reasons for including these other health problems, and they appear somewhat superfluous to the study).

564 – This entire section on other health problems detracts from the primary focus of your research. A focus on SBE here, with more detailed information, would better serve the purpose of the paper. Side data such as this is often collected, which could be another paper, or could be the groundwork for further studies, but a brief mention, rather than an entire section would be better suited.

Reviewer #3: INTRODUCTION

Need for making a stronger case for the study.

(1) Structure (lines 74-138): This study sought to deepen understanding on the role of SBE and other health challenges in a riverine community and identify strategies to address these challenges. The Introduction focuses on the historic, political, and economic factors influencing health of people in riverine communities. Whilst this background is important, especially for readers not familiar with this context, the problem of SBE, which this sought to address, should be stated upfront (ideally in the 1st paragraph). The contextual background can follow. In order to make a stronger case for the study, authors could state the evidence gap (why is this study needed) more clearly.

METHODS

Need for more concise presentation of methods and the translation of theory into methods and activities.

(2) General comment: Kindly edit to reduce unnecessary repetitions (including but not limited to the steps, methods, theory underpinning community assembly etc.)

(3) Ethics Statement (lines 141-147): Whilst this sub-section is important, it may be more effective to begin the Methods section with the context of the study (lines 175-182) and move ethics to the end of Methods. Please include a reference to supplementary ‘COREQ’ checklist. Authors may also outline measures put in place to ensure ethical conduct of the study.

(4) Study design (lines 184-218 and 247-354): (a) According to Methods, this study was proposed and designed by the research team and used a mixed-methods research design involving quantitative, qualitative, and participatory methods. The principles of Paulo Freire’s participatory education served as the epistemological foundation for the community assembly, not the entire study. Although authors state participation and horizontal power relations in theory, the Methods do not clarify how these were implemented in practice at the different stages of the research, except for the community assembly. There is need for more critical reflection on the level of participation of community members in the study beyond providing data (e.g., development of questionnaires and topic guides, enumerators and facilitators, data analysis). For example, how did the authors achieve a horizontal educator-learner relationship, when they designed the study, collected and analysed data using software (step 1), and began the community assembly (step 2) with a presentation of their results from step 1?

(b) Kindly state the duration of the study.

(5) Building rapport with the community (lines 220-229): (a) This sub-section outlines the main engagements with the community – introduction of the project, data collection, and validation (Community Assembly) – beyond building rapport. Instead of having a separate sub-section, authors may include the first two visits to the community as Step 1 “building rapport with the community” in their study design, whilst also providing more detail what happened during these visits. The other visits are already covered in study design (lines 184-218), removing these will help reduce duplication and word count.

(b) There are many abbreviations in this sub-section. Readers are not familiar with these abbreviations. Please introduce all of them when using them the first time.

(6) Research team and reflexivity (lines 231-238): Kindly consider using initials to explain the positionality of the different researchers. Please explain which efforts were made to “prevent their personal perspectives from directly influencing data collection and analysis.”

(7) Sampling (lines 240-246): (a) The study involved different methods – cross-sectional survey (structured questionnaire), in-depth interviews, focus group discussions, participant observation, and community assembly. Authors recruited “individuals who can provide valuable insights into the research topic.” Sampling and recruitment should be reported for each method separately, including the number and characteristics of participants recruited. Study area (lines 148-173): The description of the seasons affecting people in this context. She size of the population and main economic activities (lines 358-359) may be reported in this sub-section, instead of Results, and provide more details like age profile, gender composition if available.

(b) The study involved minors, please clarify the age of maturity and explain the consent procedures for minors. Please state who facilitated informed consent procedures (e.g., initials).

(8) Step 2 (lines 306-332): The structure of the section is difficult to comprehend because of a separation between the theory of the method (lines 319-323) and how it was implemented (lines 323-328), using different terminology. The theory part (lines 308-319) could be shortened.

**Results**

-Does the analysis presented match the analysis plan?

-Are the results clearly and completely presented?

-Are the figures (Tables, Images) of sufficient quality for clarity?

Reviewer #1: I am not an expert in participatory research, So I recommend inviting an additional reviewer with more specialized knowledge to offer professional insights. In section 'Discussions', I propose the authors should discuss how their study be spread in other similar regions. Replicability and scalability may be influenced by different cultural backgrounds and geographic conditions, and these aspects should be discussed deeply.

Reviewer #2: It would be helpful if the codebook were included as an appendix, as well as MAXQDA outputs such as matrices, frequencies, etc.

Table 1 – Were all 42 participants involved in each of the data collection methods, i.e., questionnaires, SSIs, focus groups? For example, were there 42 questionnaires, 42 SSIs, and how many focus groups?

Reviewer #3: Need for better integration of quantitative and qualitative findings.

(9) Participant characteristics (lines 357-373 and lines 416-423): Authors describe the quantitative sample of 42 individuals in greater depth than the qualitative sample. More detail about in-depth interview, focus group discussion, and community assembly samples needed; the labels of quotes suggest such information were collected. The participant observation sample is missing; please add. Consider reporting participant characteristics in Methods (sampling) to avoid breaking up Results.

(10) Main health problems (lines 375-414 and lines 537-562): The presentation of common health problems in the community is disjointed and, in addition, separated by the SBE findings (Theme 1). The quantitative results “Identification of the main health problems” and qualitative findings “Theme 2: Common health issues in the community” should be integrated and presented as one area of results, providing a context in which the community encounters SBEs. In addition, have authors thought to disaggregate the responses by age, gender, or other relevant characteristics as health issues appear to be related to domestic, occupational activities among adults and play among children (Table 4)?

(11) SBEs (lines 385-414 and lines 428- 535): Similarly, the quantitative evidence on exposure to venomous animals (lines 385-414) and qualitative findings on experience of SBEs (Theme 1, lines 428- 535) should be integrated and presented as one result area.

(12) Theme 3 (lines 564-639): Consider adding sub-headings for the sub-themes. It is surprising to see that participants proposed solutions are mainly curative in response to ill-health and less often preventive, except for condom use.

**Conclusions**

-Are the conclusions supported by the data presented?

-Are the limitations of analysis clearly described?

-Do the authors discuss how these data can be helpful to advance our understanding of the topic under study?

-Is public health relevance addressed?

Reviewer #1: The manuscript uses a qualitative analysis method. Although the authors also provide some descriptive statistics based on previous surveys, overall, it is insufficient. I recommend the authors provide more supporting data as appendixes to ensure the scientific research. This includes participant information, ethical approve, questionaire, etc. Regarding limitations, the authors should discussion the limitations at the last paragraph of the section 'Discussions', including the issues encountered during sample selection and data analysis. Additionally, targeted and actionable suggestions should be offered as references for similar research. This study aligns with the sustainable development goals proposed by UNESCO.

Reviewer #2: 642 – The socioeconomic findings should be presented earlier at the beginning of your data findings since this data provides context for your health findings. It does not belong in the discussion section, other than as a framing.

Again, the superfluous health data in the discussion section seems unnecessary and inappropriate.

656-664 – This ten-year reduction in SBEs is a very interesting finding. I wish there was more discussion of this change in cultural knowledge, especially implemented into the research design, with clearer data analysis of how and why this change occurred within this specific community.

I am disappointed that there is not more data collection and analysis presented on the barriers to resolving SBEs within the community. Participants provide solutions, but the likelihood of these solutions being implemented by the government are not adequately discussed. It would have been helpful if this study included all stakeholders, therefore providing better value.

742 – I am not left with the impression that “this study provides valuable insights” other than the 10-year change in cultural knowledge and practice. Please go into detail about the specific valuable insights.

753 – Decentralizing is important, but how would that work for this community specifically?

758 – I agree that community participation is a necessity, but those in control of the government apparatus need to hear the community directly, and require intermediaries since marginalized people are at a power disadvantage. This study would have been a perfect opportunity to mediate between community, health officials, and government officials, while advocating for the community.

Reviewer #3: DISCUSSION

Need for greater focus on answering the main research question.

(13) Main findings (lines 643-653): Authors begin the Discussion with a summary of participants’ sociodemographic data. However, it may be more effective to link this paragraph on the research question, providing a summary of key findings in response to the main and specific objectives. In addition, there is a disconnect between the sub-heading “River’s identity, subsistence, and invisibility” and the content; this paragraph may not need a sub-heading at all.

(14) Coherence: Reports of the importance of SBE in the riverine community are inconsistent. On the one hand, accidents involving venomous animals are perceived as a severe health problem (line 430) and ranked as the priority number one health issue in the community (Table 2). On the other hand, participants reported SBEs to be rare events (line 461), with the reported cases dating back over 10 years (lines 656-657). A discussion of these contradictory results is needed, including the potential of social desirability bias, given authors’ interest in SBEs.

CONCLUSION

(15) Conclusion (lines 742-747): The messages in the Conclusion should be focused on the research aim and data. The study generated evidence on the diversity and priorities of health issues among the riverine community as well as their proposed solutions to achieve good health. The summary of evidence in Conclusion may be misleading since “health conditions of a riverine community in the Brazilian Amazon” and “barriers to healthcare access” are not within the scope of the study.

(16) In addition, authors highlight the importance of “participatory research”, “community participation in project development”, “interdisciplinary approach”, and “methods that promote the agency of riverine communities”. Whilst not disputing the importance of participatory methods for health policy and programming, this study was not designed to assess the effectiveness or meaning of such methods. Authors may consider a critical reflection on methods in a separate manuscript if such data were available.

(17) Authors promote “invisibility” as a theme in the Title, Introduction, Discussion, and Conclusion of the manuscript, whilst “invisibility” does not explicitly feature as a theme in Results. This is not to dispute that the health needs and health concerns of the riverine population may be invisible in policy and programmes, generating the key message inductively from the data can help strengthen the manuscript. Authors may consider changing the title, dropping “Turning the invisible into visible”, and, instead, highlight participants’ request for structural, capacity, and medical support to be able to better manage their health and a health system strengthening approach to addressing SBE and other health issues.

**Editorial and Data Presentation Modifications?**

Reviewer #1: Based on the aforementioned suggestions, I recommend a major revision. The authors should address the issues outlined above appropriately.

Reviewer #2: 135: Sentence on study objective suffers from clarity issues.

144: Participants “were” provided a sheet of information on the study (?)

Were participants all literate?

220: The section on rapport building is rather anemic. Only the first visit was solely for building rapport, the others were data collection. What value did the researchers bring to the community on the first visit to build rapport?

Reviewer #3: Abstract

Please separate Methods and Results (two paragraphs). Kindly state the duration of the study, number of participants, and data collection methods. More details on the Results and less emphasis on Background and Conclusions.

**Summary and General Comments**

Reviewer #1: This study focuses on the issue of snake bites in remote regions of the Amazon, addressing a neglected public health problem. Given the high incidence of snake bites in this area, the manuscript holds significant importance in the field of neglected tropical diseases. The solutions proposed in the study, such as first aid training, access to satellite phones, and infrastructure improvements, are practical and can directly inform local health policy interventions. However, I believe the authors still need to address the following issues to enhance their chances of publication.

Existing Issues:

1. It is recommended that the authors conduct a more in-depth discussion of the descriptive statistics of the questionnaires and interviewees. This should include how differences in occupation, age, gender, and culture affect attitudes toward health issues, which would help improve the design of community services.

2. As mentioned above, based on the study results, can the authors establish several priority actions to illustrate how the findings can be extended to other tropical communities?

3. Additionally, why did the authors choose subjective sampling instead of stratified or mixed sampling? The authors need to provide sufficient justification for this choice, as it directly affects subsequent research results. For example, subjective sampling may exclude more marginalized groups in the community or those less familiar with the health system.

Suggestions to authors:

1. Enhance the Depth of Quantitative Analysis:In the subsection "Step 1 - Community baseline assessment," page 11, third paragraph, only simple descriptive statistical analysis of the collected questionnaire data is currently presented. It is recommended to add further quantitative analysis after this paragraph, using multivariate analysis methods (such as linear regression or logistic regression) to explore the correlation between different demographic characteristics (e.g., gender, age, occupation) and various health issues. This will make the conclusions more reliable and facilitate the generalization of results to other similar communities.

2. It is suggested to add a follow-up plan based on the existing research framework to observe the long-term effects of the proposed interventions (such as first aid training and infrastructure improvements). By collecting data on changes in community health status before and after the interventions, the actual impact of these measures can be better evaluated, thus providing stronger evidence for future public health policies.

3. Expand the Discussion on Environmental Factors: In the "Discussion" section, page 34, second paragraph (the paragraph starting with "The frequency of snake encounters is high"), after discussing the frequency of community members' contact with venomous snakes, it is recommended to further elaborate on the impact of climate change and environmental degradation on snake behavior and other health issues. Relevant studies can be cited to illustrate how climate change increases the range of snake activity, thereby raising the risk of snake bites. Additionally, other environmental issues (such as deforestation and deterioration of river water quality) and their potential impacts on infectious and occupational diseases can be discussed. This can elevate the study's discussion to the context of global environmental changes, enhancing the paper's cutting-edge nature.

Reviewer #2: This study provides value as a case study, especially in terms of the 10-year difference observed in SBE cultural knowledge and adaptation. I would recommend shifting the framing of the writing to highlight this finding. As outlined, I found the data on other health problems to be distracting from the focus of the study.

Reviewer #3: Researchers proposed this study to establish the importance of and solutions to snake bite envenoming among riverine communities in Brazil, using a mixed-methods research design involving quantitative, qualitative, and participatory methods. They engaged community members on several occasions during the study to build rapport, collect data, and develop solutions. Researchers made an effort to engage with the views of participants from the community, including widening the research focus to include other conditions, but evidence for their consistent involvement in horizontal power relations is limited. The Results provide opportunities for action. However, their presentation and discussion may be integrated and focussed on the research question. Given the active role researchers have in a study like this, active language and author initials may allow to reader to better understand who carried out what activity.

This review identifies opportunities to strengthen this manuscript like making a stronger case for the study (Introduction); presenting methods more concisely, including how theory was translated into methods and activities (Methods); integrating quantitative and qualitative findings (Results); and strengthening the Discussion to focus on the research question (Discussion/Conclusion).

PLOS authors have the option to publish the peer review history of their article (what does this mean?). If published, this will include your full peer review and any attached files.

Reviewer #1: No

Reviewer #2: No

Reviewer #3: No

**Figure resubmission:**
---

## [Decision Letter · Decision Letter 1]

6 Jan 2025

PNTD-D-24-01262R1

Participatory research towards the control of snakebite envenoming and other illnesses in a riverine community of the Western Brazilian Amazon

Dear Dr. Murta,

Thank you for submitting your manuscript to PLOS Neglected Tropical Diseases. After careful consideration, we feel that it has merit but does not fully meet PLOS Neglected Tropical Diseases's publication criteria as it currently stands. Therefore, we invite you to submit a revised version of the manuscript that addresses the points raised during the review process.

Please submit your revised manuscript within 60 days. If you will need more time than this to complete your revisions, please reply to this message or contact the journal office at plosntds@plos.org. Please include the following items when submitting your revised manuscript:

We look forward to receiving your revised manuscript.

Kind regards,

Soumyadeep Bhaumik, MBBS, M.Sc, PhD

Academic Editor

José María Gutiérrez

Section Editor

Shaden Kamhawi

co-Editor-in-Chief

Paul Brindley

co-Editor-in-Chief

**Additional Editor Comments:**

Thank you for revising the manuscript which has majorly improved the article. However please review the comments by the two peer-reviewers and make changes, if necessary. Having the critical discussion might be preferred as it enables readers to make their own interpretation. However it is essential to ahve a very clear flow of article . It might be a separate sub-section in the discussion. Please also ensure all recommendations and conclusions are based on the current paper and not broad and generic in nature. It will enhance the utility of the paper.

**Reviewers' Comments:**

Reviewer's Responses to Questions

**Key Review Criteria Required for Acceptance?**

**Methods**

-Are the objectives of the study clearly articulated with a clear testable hypothesis stated?

-Is the study design appropriate to address the stated objectives?

-Is the population clearly described and appropriate for the hypothesis being tested?

-Is the sample size sufficient to ensure adequate power to address the hypothesis being tested?

-Were correct statistical analysis used to support conclusions?

-Are there concerns about ethical or regulatory requirements being met?

Reviewer #1: Issues on old manuscripts have been resolved.

Reviewer #3: This version of the manuscript is clearer, adding suggestions for minor corrections.

Introduction: The flow is improved. However, the added hypothesis seems to focus on the effectiveness or value of a participatory research approach, not the views of a riverine community regarding their health issues and healthcare needs. I would advise against mixing methodological and empirical findings. Authors may consider evaluating the methodological approach, offering a critical reflection in a separate manuscript.

Methods: Information requested has been added and methods clarified.

Sampling: It seems a stratified sampling approach was used for FGDs. The sampling strategy for IDIs and the IDI sample composition remain unclear. Who was invited and who took part (age, gender)?

Research team: Authors have introduced the research team. The team is rather big (compared to the study). Kindly clarify in the text the role of each team member in the study. This can be through initials throughout the Methods chapter.

**Results**

-Does the analysis presented match the analysis plan?

-Are the results clearly and completely presented?

-Are the figures (Tables, Images) of sufficient quality for clarity?

Reviewer #1: Please use data visualization tools to present the table content in a more intuitive and informative way.

Reviewer #3: Presentation/ structure: Whilst integration of results was recommended, authors decided to keep the quantitative and qualitative findings separate. Sub-themes improve structure.

Quotes: The quotes are well labelled. Changes may be high that participants can be identified because of the detailed information given and the small size of the community (64 people). Can identities be better protected?

**Conclusions**

-Are the conclusions supported by the data presented?

-Are the limitations of analysis clearly described?

-Do the authors discuss how these data can be helpful to advance our understanding of the topic under study?

-Is public health relevance addressed?

Reviewer #1: The revised manuscript is a major improvement over the old version, and has basically solved most of the problems I raised previously, and I recommend that it be adopted for publication.

Reviewer #3: Discussion: Authors have made changes. A introduction paragraph linking back to the research gap and question is still needed.

Limitations: More critical engagement with the drop in the number of participants from survey (n=42) to community meeting (n=20) is needed. The study was researcher-led whilst aiming at actively involving participants. Any recommendations for future research how to mitigate against drop out?

Conclusion: As stated before, the messages in the Conclusion should be focused on the research aim and data – relative importance of SBE and improved access to healthcare in the riverine community, not the participatory methods. Kindly highlight the empirical results and new knowledge gained from the study. A critical evaluation of the method should be offered in a separate manuscript.

**Editorial and Data Presentation Modifications?**

Reviewer #1: Accept.

Reviewer #3: Minor Revision

**Summary and General Comments**

Reviewer #1: The revised manuscript is a major improvement over the old version, and has basically solved most of the problems I raised previously, and I recommend that it be adopted for publication.

However, the following issues should be noted in the subsequent publication.

1. the use of data visualization tools to present the contents of the tables in a more intuitive and informative way.

2. the consistency of language presentation in the manuscript, avoiding colloquial expressions and standardizing American or British spelling.

3. ensure that the research is cutting-edge and authoritative, ensure that more than 75% of the references come from Clarivate and Scopus databases, and do not cite withdrawn papers.

Reviewer #3: The manuscript has been improved. Authors should focus on the empirical results. The evaluation of the methods should be reported in a separate manuscript and not mixed with the Results.

PLOS authors have the option to publish the peer review history of their article (what does this mean?). If published, this will include your full peer review and any attached files.

Reviewer #1: **Yes: **Ao Jinghui

Reviewer #3: No

**Figure resubmission:**
---

## [Editor Report · Decision Letter 2]

14 Jan 2025

Dear Dr Murta,

We are pleased to inform you that your manuscript 'Participatory research towards the control of snakebite envenoming and other illnesses in a riverine community of the Western Brazilian Amazon' has been provisionally accepted for publication in PLOS Neglected Tropical Diseases.

Best regards,

Soumyadeep Bhaumik, MBBS, M.Sc, PhD

Academic Editor

José María Gutiérrez

Section Editor

Shaden Kamhawi

co-Editor-in-Chief

Paul Brindley

co-Editor-in-Chief

---

## [Editor Report · Acceptance letter]

17 Jan 2025

Dear Dr. Murta,

We are delighted to inform you that your manuscript, "Participatory research towards the control of snakebite envenoming and other illnesses in a riverine community of the Western Brazilian Amazon," has been formally accepted for publication in PLOS Neglected Tropical Diseases.

Best regards,

Shaden Kamhawi

co-Editor-in-Chief

Paul Brindley

co-Editor-in-Chief
